# Significance of Mitochondrial Dysfunction in the Progression of Multiple Sclerosis

**DOI:** 10.3390/ijms232112725

**Published:** 2022-10-22

**Authors:** Alexander V. Blagov, Vasily N. Sukhorukov, Alexander N. Orekhov, Margarita A. Sazonova, Alexandra A. Melnichenko

**Affiliations:** 1Laboratory of Angiopathology, Institute of General Pathology and Pathophysiology, 8 Baltiiskaya Street, 125315 Moscow, Russia; 2Institute for Atherosclerosis Research, Osennyaya Street 4-1-207, 121609 Moscow, Russia

**Keywords:** multiple sclerosis, mitochondrial dysfunction, neurons

## Abstract

The prevalence of multiple sclerosis and the complexity of its etiology and pathogenesis require further study of the factors underlying the progression of this disease. The prominent role of mitochondria in neurons makes this organelle a vulnerable target for CNS diseases. The purpose of this review is to consider the role of mitochondrial dysfunction in the pathogenesis of multiple sclerosis, as well as to propose new promising therapeutic strategies aimed at restoring mitochondrial function in multiple sclerosis.

## 1. Introduction

Multiple sclerosis (MS) is the most common chronic inflammatory neurological disease: according to epidemiological data, 2.8 million people in the world were diagnosed with MS by the end of 2020, and its prevalence increased in all regions compared to the statistics for 2013 [1]. MS is a multifocal demyelinating disease with progressive neurodegeneration associated with an autoimmune response to autoantigens [2]. The average age of diagnosis of MS disease is 32 years, which characterizes MS as a disease of the “young” in contrast to many other neurological diseases [1]. The most typical situation, which occurs 10–20 years after the onset of the disease in many patients, is the development of a “progressive” clinical course of MS, in which there is a steady worsening of symptoms with possible periods of remission, and which eventually manifests itself in the form of impaired mobility and cognitive functions. [3]. The etiology of MS is still unclear. Possible genetic risk factors for the development of MS, as well as environmental factors, have been identified. Most likely, the occurrence of an autoimmune reaction in MS is not due to the action of any specific factor, but the cumulative interaction between several factors. As a result of genome-wide association studies, more than 200 gene variants associated with the risk of developing MS have been identified, most of which determine immune interactions [3]. Environmental risk factors include obesity, smoking, mononucleosis (caused by Epstein-Barr virus infection), and components of the gut microbiome [3]. The main method used for the diagnosis of MS is MRI, which allows you to track the state of the central nervous system in vivo. To confirm the diagnosis of MS, two or more MRI-detected lesions at different locations in the CNS are required, consistent with two or more clinical episodes that appear over time [2]. Other methods used in the diagnosis of MS are the study of cerebrospinal fluid (CSF), as well as tests for intrathecal synthesis of immunoglobulin G [4]. Unfortunately, at the moment there are no MS biomarkers contained in blood serum with high efficiency and sensitivity for reliable detection of the disease, which would allow much faster diagnosis [3]. To date, there are no developed therapeutic agents that can completely cure MS, however, available drugs can significantly reduce the progression of the disease and alleviate the condition of patients [5]. MS is treated with drugs that either modulate or suppress immune function. Glatiramer acetate, the first drug developed for the treatment of MS, increases the production of anti-inflammatory cytokines and reduces the production of pro-inflammatory [6]. The monoclonal antibody drug ocrelizumab targets CD20-bearing B-lymphocytes, which suppresses the humoral autoimmune response [7]. Another antibody, natalizumab, inhibits α4β1 integrin adhesion molecules, which are expressed on the surface of lymphocytes and promotes the migration of immune cells through the endothelium to the CNS [5]. Fingolimod inhibits the migration of lymphocytes from secondary lymphoid organs, which blocks the infiltration of autoreactive lymphocytes into the CNS [5]. Despite the positive therapeutic effects, the drugs used have a number of severe side effects, which limits their use [5]. A better understanding of the processes underlying the pathogenesis of MS could lead to the discovery of new therapeutic targets and biomarkers, which will be a new step in the development of new types of drugs against MS and new diagnostic tests. The detection of pathologies that lie at the level of intracellular functioning is an important issue for a better understanding of the pathogenesis of MS. In this review, we will focus on the consideration of mitochondria as a pathological link, the disruption of which leads to the clinical manifestations of the disease. The mitochondrion, as will be described below, plays an important role in the functioning of the cell. At the same time, it has been noted that mitochondrial dysfunction is associated with the development of a number of chronic diseases, including CNS diseases [8,9].

## 2. The Pathogenesis of Multiple Sclerosis

### 2.1. Types of Multiple Sclerosis

The two main pathological features of MS are inflammation followed by demyelination and proliferation of astroglia (gliosis) with concomitant neurodegeneration [5]. Tissue damage in multiple sclerosis is limited to the central nervous system (CNS) without affecting the peripheral nervous system [5]. According to clinical manifestations, multiple sclerosis can proceed in two ways: relapsing or progressive [10]. Most often, the onset of the disease is associated with a relapsing form of multiple sclerosis (RRMS), which manifests itself in the occurrence of discrete episodes of neurological dysfunction, followed by partial, complete remission or its absence. Over time, the frequency of relapses most often decreases, but at the same time there is a gradual deterioration, leading to the continuous progression of the disease—a stage called secondary progressive MS (SPMS). The form of MS in which the disease progresses from the very beginning is called primary progressive multiple sclerosis (PPMS). However, despite the differences, all clinical forms of MS most likely reflect a common pathological process. Although inflammation is most often associated with relapses of the disease, and neurodegeneration with the progression stage, it is now recognized that both pathologies are observed in all patients with MS throughout the course of the disease [5].

### 2.2. Initiation of an Autoimmune Reaction

The pathogenesis of MS is based on an autoimmune reaction directed against CNS antigens, which is carried out by activated CD4+ myelin-reactive T cells with the possible involvement of B cells. It is believed that the basis for the development of an autoimmune reaction in MS is a violation of tolerance to myelin and other CNS antigens, which leads to persistent peripheral activation of autoreactive T-lymphocytes [11]. In people with genetic risk factors for MS, this loss of immune tolerance may be caused by an antigen from the environment, primarily an infectious agent such as a virus. Infection can cause additional T-cell activation or result in the release of self-antigens due to cellular damage, which can subsequently set a precedent for T-lymphocyte activation as a result of cross-reactivity between an endogenous protein (e.g., myelin basic protein) and a pathogenic exogenous protein (viral antigen) [12].

### 2.3. Triggering Inflammation in the CNS

Once activated in the periphery, myelin-reactive T-lymphocytes can migrate across the blood–brain barrier (BBB). The transmigration process is carried out through the interaction between the VLA-4 receptor, present on the surface of T-lymphocytes, and the VCAM-1 receptor, which is located on capillary endothelial cells. The transmigration process is further aided by the activation of various adhesion molecules, chemokines, and matrix metalloproteinases (MMPs) [13]. Once autoreactive, peripherally activated T lymphocytes have crossed the BBB into the CNS, they may be reactivated by encountering autoantigenic peptides in the brain parenchyma that bind to MHC II molecules that are expressed by local antigen-presenting cells (dendritic cells, macrophages, and B-cells). cells). As a result of the antigen presentation process, an inflammatory response is triggered, which leads to the release of cytokines and chemokines, as well as the recruitment of additional immune cells to the inflammation site with additional activation of microglia, which ultimately leads to damage to myelin [14].

### 2.4. Roles of Immune Cells

Local inflammation and demyelination lead to the spread of sequestered myelin autoantigens, which are an additional target for autoreactive T-lymphocytes [15,16,17]. Activation of microglial cells, which are CNS resident macrophages, is associated with the development of persistent inflammation even in the absence of further infiltration of exogenous lymphocytes [14]. Based on studies on animal models of MS, it can be assumed that the key role in the development of an autoimmune reaction in MS belongs to the subpopulation of Th 1 CD4 + T-lymphocytes, as well as the subpopulation of Th 17 CD4 + T-lymphocytes expressing IL -17, a cytokine involved in pathogenesis of many autoimmune diseases [13]. In addition to demyelination, autoreactive T-lymphocytes activated by microglial cells or antibodies can cause permanent damage to axons [18]. A subpopulation of Th 2 CD4 + T lymphocytes, secreting the anti-inflammatory cytokines IL 4, IL 5, and IL 10, has been suggested to act as inhibitors of autoimmune inflammation, limiting Th1-mediated damage [19]. In addition, CD8+ T-lymphocytes are also directly involved in the development of MS and can induce axonal pathology by direct damage to cells (neurons, oligodendrocytes) that express the MHCI complex with autoantigen on their surface [20]. The involvement of B cells, which is determined primarily by the secretion of autoreactive antibodies and the presentation of autoantigen to T lymphocytes, has also been confirmed in the pathogenesis of MS [20]. The summarized scheme of MS pathogenesis is depicted in Figure 1.

## 3. The Role of Mitochondria in the Functioning of Cells

Mitochondrial dysfunction may be associated with the development of many chronic diseases, including MS. This is due to the vital functions that the mitochondrion performs in the cell and, accordingly, the disruption of these functions leads to the development of pathologies. The main functions of mitochondria in the cell are the production of ATP, participation in the metabolism of many vital compounds, the role in the generation of reactive oxygen species (ROS), which are important signaling molecules, as well as the initiation of the internal pathway of apoptosis and the antiviral immune response.

### 3.1. ATP Production

ATP synthesis is the main vital function performed by mitochondria. ATP is the main energy unit used to carry out all the processes that require energy in the body. ATP is produced during energy metabolism as a result of the oxidation of organic substrates (primarily glucose). In the course of oxidation during the Krebs cycle occurring in the mitochondrial matrix, electrons are formed, which, in the form associated with carriers (NADH and FADH 2), are transported to the inner mitochondrial membrane, where they are sequentially transferred through an electron transport chain consisting of 4 protein complexes [21]. As a result of successive redox reactions, a proton gradient is generated, which is used by the ATP synthase enzyme to pump protons from the intermembrane space into the mitochondrial matrix, resulting in the formation of ATP from ADP, and the process underlying it is called oxidative phosphorylation [22]. Neurons are characterized by a high need for energy, which is necessary for the implementation of high-energy processes that underlie the conduction of a nerve impulse. Because of this, neurons are highly dependent on the proper functioning of the mitochondrial network and highly vulnerable to mitochondrial dysfunction. In connection with the development of mitochondrial dysfunction in MS, insufficient ATP generation occurs, which is reflected in a decrease in the functional activity of neurons and leads to neurogeneration, which will be described in more detail in the next section.

### 3.2. ROS Generation

ETC complexes I and III also generate ROS during electron transfer, including oxygen radicals and hydrogen peroxide [21]. ROS are by-products of oxidative phosphorylation, yet they are important cell signaling molecules. The content of ROS in the cell is important for cellular homeostasis, it is believed that under normal physiological conditions, the production of ROS is 2% of the total amount of oxygen consumed by mitochondria [23]. A change in the level of ROS in both directions negatively affects the cells: a lower level, as well as a higher level, can have harmful consequences. Thus, a low level of ROS cannot ensure the working functioning of cells through the regulation of a cascade of biochemical reactions [24]. In the case of high concentrations of ROS, they change their role from signal to pathological and act as destroyers of macromolecules: proteins, lipids, and nucleic acids [24]. The state of high content and increased generation of ROS in the cell is called oxidative stress, which contributes to cell death and the development of inflammation, and also underlies the pathogenesis of various diseases, including MS. Neurons have an increased vulnerability to oxidative damage due to the high rate of oxygen supply to ensure rapid energy metabolism, and also due to the large variety of polyunsaturated fatty acids, which are one of the main targets for ROS [25]. At normal levels, mitochondrial ROS act on homeostatic signaling pathways, thus controlling cell proliferation and differentiation, and are able to participate in adaptive stress signaling pathways such as hypoxia [26,27].

### 3.3. Metabolism of Organic and Inorganic Compounds

In addition to direct energy metabolism associated with the synthesis of ATP, other important metabolic processes take place in the mitochondria. In all cell types, mitochondria are the main cellular source of NADH, and they also carry out part of the reactions of pyrimidine biosynthesis and lipid metabolism, including the fatty acid β-oxidation pathway, as a result of which long-chain fatty acids are transformed into acyl-CoA [21]. Mitochondria also regulate cellular levels of other metabolites, including amino acids and cofactors of various regulatory enzymes such as histone deacetylases, whose function is to modify chromatin [21]. Mitochondria also play a significant role in metal metabolism through the production of heme and iron-sulfur clusters, which are essential structures for the functioning of the oxygen carrier, hemoglobin [28]. An additional function of mitochondria is based on the homeostasis of Ca^2+^ ions through the spatiotemporal distribution of this signaling mediator as a result of the process of buffering the supply of Ca^2+^ ions from the plasma membrane and endoplasmic reticulum (ER) [29]. In neurons, the effect of mitochondria on the modulation of Ca^2+^ flux is necessary to control the release of neurotransmitters, neurogenesis, and neuronal plasticity. In addition, intermediate metabolites formed during the Krebs cycle serve as building materials necessary for the production of neurotransmitters such as GABA and glutamate [30]. In MS, the resulting imbalance of ion exchange in neurons leads to the activation of apoptosis.

### 3.4. Mitochondrial-Mediated Apoptosis

Mitochondria are direct participants in the internal pathway of apoptosis, during which the death of malignant or damaged cells occurs. The mitochondrial pathway is initiated in response to cell death stimuli, which include DNA damage, exposure to chemotherapeutic agents, serum depletion, and UV radiation [31]. The mitochondrion, in addition to being the site where anti-apoptotic and pro-apoptotic proteins interact, is also a source of outgoing signals that initiate the activation of caspases, the main destructive enzymes during apoptosis, through the action of various mechanisms [31]. Thus, a number of mitochondrial proteins, which under normal conditions can perform other functions, initiate the action of caspases during apoptosis. For example, cytochrome c is a necessary component of the apoptosome complex that serves to activate caspase-9. After leaving the mitochondria, the mitochondrial protein Smac, which is another caspase activator, and Omi are able to bind to apoptosis inhibitory proteins (IAP) and, as a result, reduce their inhibitory effect on caspase activity [31]. Mitochondrial-dependent apoptosis is a direct pathway for neuronal death in neurodegenerative diseases, including MS.

## 4. Mitochondrial Dysfunction in the Pathogenesis of Multiple Sclerosis

### 4.1. Mitochondrial Abnormalities in MS

A number of mitochondrial abnormalities have been identified based on studies of samples from patients with MS, as well as on the study of mouse models of experimental autoimmune encephalitis. These anomalies included: increased mutations in mitochondrial DNA, impaired expression of mitochondrial genes, defective activity of mitochondrial enzymes, reduced ability to repair mitochondrial DNA, as well as disturbances in the balance of mitochondrial dynamics and changes in the content of metabolites of energy metabolism in the cell [32,33]. All these processes underlie the development of mitochondrial dysfunction, the occurrence of which is a proven fact in MS [33]. Mitochondrial dysfunction has a particularly strong effect on neurons due to their high dependence on the proper functioning of mitochondria as was discussed in Section 3.1. 

### 4.2. The Role of Oxidative Stress in the Pathogenesis of MS

Inflammation is a pathogenic reaction observed from the very beginning of the development of MS in the body from the moment of the emergence of autoreactive forms of lymphocytes in peripheral tissues. Penetrating through the BBB, immune cells initiate the development of neuroinflammation, which becomes chronic. One of the fundamental factors leading to an increase in inflammation is oxidative stress, which is formed as a result of the production of ROS in the focus of inflammation. There is evidence that oxidative stress is an important component of the pathogenesis of MS. Thus, macrophages and microglia are known to produce ROS during phagocytosis of myelin in the white matter [34]. Another evidence of the spread of oxidative stress in inflammatory demyelinating lesions in MS is the detection of markers of lipid peroxidation and protein peroxides (protein carbonyls), as well as the presence of 8-hydroxydeoxyguanosine, which is an oxidative marker of DNA damage [33]. Oxidative damage to mitochondrial DNA and disruption of the activity of mitochondrial enzyme complexes in MS lesions leads to mitochondrial dysfunction, as a result of which the process of oxidative phosphorylation is disrupted and ROS production by mitochondria increases [33]. As a result, oxidative stress occurs already inside neurons and glial cells, which leads to damage to intracellular proteins, lipids, and DNA, and the appearance of secondary metabolites that can play the role of additional autoantigens. In addition, ROS directly damage the myelin sheath, promoting the release of new autoantigenic particles, which causes an increase in autoimmune inflammation and subsequent damage to neuronal structures [35,36]. Restoration of mitochondrial DNA is one of the key ways to protect cells from the development of oxidative stress. It is known that this process is carried out predominantly by base excision repair (BER), which includes the work of removing damaged bases with the participation of the enzymes DNA glycosylase and apurine endonuclease and subsequent repair of the nucleic acid strand due to the action of DNA polymerase γ and DNA ligase. The reasons for impaired mitochondrial DNA recovery in MS are not completely clear; however, from ongoing studies on models of CNS diseases, it is known that the expression and activity of BER enzymes can change, and both increase (apurine endonuclease) and decrease (DNA glycosylase), which leads to an imbalance in the process of repair of mitochondrial DNA [37]. Since oligodendrocytes are the generators of the myelin sheath, their damage is one of the decisive factors in the progression of MS. Unlike neurons and astrocytes, oligodendrocytes are more sensitive to oxidative damage due to reduced antioxidant protection, which makes them a convenient target for ROS damage and subsequent axonal demyelination [33].

### 4.3. Disruption of Bioenergetics in the Pathogenesis of MS

The emerging mitochondrial dysfunction, in addition to additional generation of ROS, leads to the disruption of oxidative phosphorylation, as well as other upstream metabolic processes involving mitochondria. As a result of disturbance of metabolic pathways, an imbalance of neurotrophic substances for neurons and oligodendrocytes occurs, which leads to increased demyelination of axons [25]. Thus, N-acetylaspartate is a mitochondrial metabolite, as well as an indirect substrate of oligodendrocytes for the production of myelin (after decomposition into acetate and aspartate). Decreased N-acetylaspartate uptake from dysfunctional mitochondria has been associated with lower levels of acetate in the parietal and motor cortex in post-mortem tissue samples from MS patients [38]. Another consequence of pronounced dysfunction of mitochondria in neurons is the occurrence of excitotoxicity due to impaired metabolism of neurotransmitters, which leads to impaired neuronal function and initiation of apoptosis [25]. A common result of mitochondrial dysfunction in neurons is a decrease in the efficiency of energy production, resulting in an imbalance between energy production and consumption. As a result, energy starvation leads to a state of inability to provide the required level of energy in demyelinated axons [25]. An additional decrease in ATPase activity leads to an imbalance of ions in axons, caused primarily by an increased influx of calcium ions into axons, which leads to the launch of pathways that initiate cell death [39].

### 4.4. Disruption of Mitochondrial Dynamics in MS

Correctly coordinated work of mitochondrial dynamics processes is the cornerstone for maintaining the vital activity of cells, especially those with a highly polarized and branched structure, such as neurons. However, in pathological conditions, impaired mitochondrial dynamics is a signal that enhances the negative consequences for the cell caused by mitochondrial dysfunction [40]. It has been shown that both anterograde and retrograde transport of mitochondria in axons are inhibited in MS, which is caused by a decrease in the expression of motor proteins (kinesins) in neurons [41]. Disruption of anterograde transport prevents the delivery of new mitochondria from the neuron body to the axon, which further exacerbates energy starvation in axons. Disruption of retrograde transport leads to the accumulation of old non-functioning mitochondria in the axon, which are an additional source of ROS and, accordingly, oxidative stress. At the same time, in MS patients, there is an increase in the level of mitophagy in neurons, although it remains unclear whether this process is associated with pathogenesis or has a compensatory mechanism [42]. The scheme of MS pathogenesis based on mitochondrial disfunction is depicted in Figure 2.

## 5. Potential Therapeutic Strategies to Restore Mitochondrial Function

Taking into account the role of mitochondrial dysfunction in the pathogenesis of MS, as shown above, it is worth considering possible ways to restore the efficiency of the mitochondrial system in neurons and glial cells, which can be applied in practical terms in MS therapy. Potentially, this will contribute to the creation of neuroprotective therapies, which will be new and possibly more effective than currently used approaches in the treatment of MS. One of these areas is the artificial repair of mitochondrial DNA. Thus, a strategy was developed for the delivery of DNA repair proteins to mitochondria. Enhanced repair of mitochondrial DNA in oligodendrocytes provided protection against apoptosis induced by reactive oxygen species and cytokines [43]. The second therapeutic strategy could be the use of ion channel inhibitors, since, as noted in the previous section, ion imbalance is one of the consequences of mitochondrial dysfunction leading to axonal depletion and neuronal apoptosis. Thus, it was shown that the blocker of Na + channels tetrodotoxin prevents the decrease in the concentration of ATP while protecting axons from NO-induced damage [44]. Additionally, another Na+-channel blocker, phenytoin, was found to have a protective effect on maintaining axonal function in animal models of experimental autoimmune encephalitis [45]. The third potential strategy is based on the use of mitochondria-targeted antioxidants, such as: MitoQ, SS31, CART, since oxidative stress is an initiating and concomitant factor in mitochondrial dysfunction [46,47]. A study [48] showed the neuroprotective effect of CART with the preservation of mitochondrial function and the prevention of energy insufficiency. For the targeted treatment of multiple sclerosis, the use of antioxidants such as alpha-lipoic acid and coenzyme Q10 is being studied. According to the results of the second phase of the clinical study of the use of alpha-lipoic acid in the treatment of MS, it was found that this compound slows down atrophy of the entire brain [49]. In a Phase I/II clinical trial of the synthetic analog of coenzyme Q10, idebenone, this compound was shown to be therapeutically safe, but was not effective in reducing disease progression [50]. A truly innovative therapy is based on the transplantation of healthy mitochondria into cells with dysfunctional mitochondria. However, it is too early to talk about clinical efficacy in the treatment of patients with MS by using this method [51].

## 6. Discussion

As shown in the previous section, scientists are exploring various options for creating new therapeutic agents aimed at restoring mitochondrial function in nerve cells. Some of these developments have even successfully proven themselves at the stages of clinical trials, which indicates a high therapeutic potential in this area. But despite this, at the moment there are no registered drugs aimed at restoring mitochondrial function in MS. Perhaps this may be due to the fact that MS is a multifactorial disease and exposure to one pathological factor, for example, oxidative stress or disruption of ion channels, does not lead to the desired therapeutic effect, since in different patients the contribution of pathological processes to the development of the disease can significantly differ. From our point of view, drugs that act on protein regulators of mitochondrial synthesis and expression of ETC components will have the greatest efficiency. An example of such regulatory proteins can be PGC-1α, the modulation of whose activity was described in the study [52]. The advantage of this method is the possibility of a complex effect on the restoration of the function of the entire intracellular network of mitochondria in the initial stages of mitochondrial biogenesis. Thus, this will allow to deal directly with mitochondrial dysfunction, and not with its consequences, and if mitochondrial dysfunction is indeed an important factor in the pathogenesis of most patients with MS, then this strategy promises to be effective. At the same time, there is a need to develop rapid and efficient diagnostic tests to determine the degree of MS progression in patient samples. Perhaps mitochondria can become an assistant in this case as well. As shown in a study [53], the concentrations of metabolites in the cerebrospinal fluid differ in MS patients and healthy people, which reflects mitochondrial dysfunction and, accordingly, the pathological process. A more detailed study of changes in the dynamics of metabolites in response to mitochondrial dysfunction may be the key to the development of new diagnostic tests. Research is also underway to identify specific biomarkers. So, in the study [54], the mitochondrial biomarker GDF-15 showed a positive correlation depending on the severity of MS. Moreover, in a clinical study [55], it was shown that the mitophagy regulatory proteins PINK1 and PARKIN have increased values in patients with MS compared with the control group. And in a clinical study [56], lactate levels were on average 3.5 higher in MS patients than in healthy controls. Additional study requires the state of mitochondria in other cell types, in addition to nerve cells. Thus, it was shown that ETC complex IV defects are manifested in axons, astrocytes, and oligodendrocytes, but are not found in microglial cells [57]. Does this mean that mitochondrial dysfunction is characteristic only of cells of the nervous system? However, it has also been found that the muscles of MS patients contain a lower concentration of oxidative phosphorylation enzymes compared to healthy people [58]. It is known that MS can be associated with mitochondrial cardiomyopathy [59], but other cell types may also be the target of MS. Understanding how the metabolism of immune cells (primarily T-helper cells) changes in response to the occurrence of an autoimmune reaction may be important for the development of therapies that affect the metabolic pathways of autoreactive lymphocytes, which will potentially prevent the spread of an autoimmune reaction. Additional study also requires how other processes of mitochondrial dynamics besides mitochondrial transport change, and how these changes, if they occur, contribute to the progression of MS. An important question is: how common are certain mitochondrial abnormalities in MS patients? The answer to this question will make it possible to identify the groups of the most frequent mitochondrial disorders in MS patients, which will contribute to more targeted development of therapeutic agents against MS.

## 7. Conclusions

Many factors underlie the initiation and progression of MS, making this disease a difficult subject to study. Inflammation and neurodegeneration are two key processes that determine the pathogenesis of MS. Penetrating into the CNS, immune cells trigger neuroinflammation as an autoimmune response to binding to myelin. The generation of ROS leads to the development of oxidative stress, which increases inflammation, leading to even greater damage to the myelin sheath and neurodegenerative processes. Oxidative stress also causes mitochondrial dysfunction, which leads to impaired neurotrophic nutrition of neurons and oligodendrocytes, development of excitotoxicity, increased axonal demyelination, and ion imbalance, which ultimately results in apoptosis of neurons and glial cells. This situation is exacerbated by impaired mitochondrial transport. Several potential strategies have been proposed to restore mitochondrial function, which include: the use of mitochondrial DNA repair proteins, ion channel inhibitors, and the use of antioxidants.

## Figures and Tables

**Figure 1 ijms-23-12725-f001:**
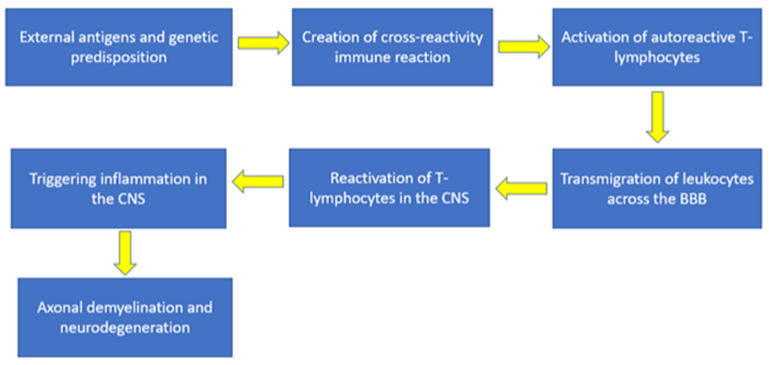
The summarized scheme of MS pathogenesis.

**Figure 2 ijms-23-12725-f002:**
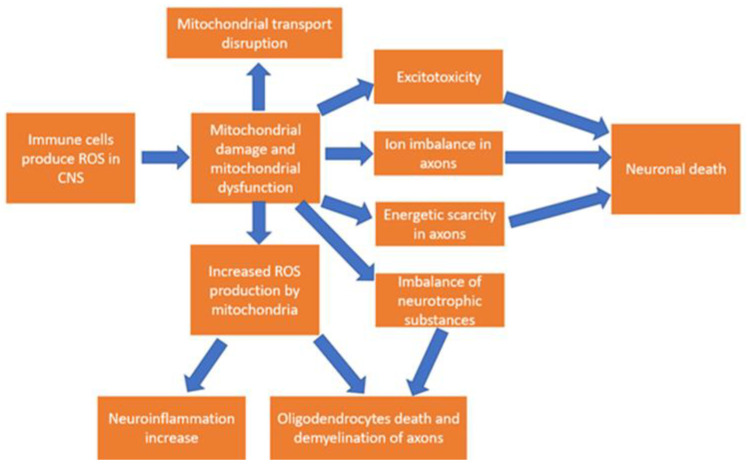
The scheme of MS pathogenesis based on mitochondrial disfunction.

## Data Availability

Not applicable.

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
