# Peer review of "Significance of Mitochondrial Dysfunction in the Progression of Multiple Sclerosis"

_ijms, 2022, doi:10.3390/ijms232112725_

Round 1
Reviewer 1 Report
The number of literature references could have been higher since this is a review article. As this manuscript is submitted now in September 2022 as a Review on a very interesting topic, it would be advisable to include in the Review any relevant recent publications that are topically related, thereby increasing the added value of this manuscript and generating interest among relevant specialists. Authors can sift sufficiently significant articles from the 163 thematically related 2020-2022 publications find in PubMed. However, in the proposed manuscript, the authors have referred to only four publications of 2020.
The manuscript should be updated.
Author Response
Response to Reviewer 1 Comments
Point 1:
he number of literature references could have been higher since this is a review article. As this manuscript is submitted now in September 2022 as a Review on a very interesting topic, it would be advisable to include in the Review any relevant recent publications that are topically related, thereby increasing the added value of this manuscript and generating interest among relevant specialists. Authors can sift sufficiently significant articles from the 163 thematically related 2020-2022 publications find in PubMed. However, in the proposed manuscript, the authors have referred to only four publications of 2020.
The manuscript should be updated.
Response 1: The references for new articles published in 2020-2022 have been added. References №: 16,17,27,32,37,40,46,47,51,52,54,55,56.
Reviewer 2 Report
Please shorten the introduction and focus on the underlying rationale for this review article.
Please also rephrase the first paragraphs on mitochondrial function. It should be, in general, clear to the readership that mitochondria are involved in, i.e., ATP biosynthesis. What is the precise role of these mechanisms in MS pathophysiology?
The different sections in the manuscript are somewhat disproportionate. The. The authors start with the main concept of the review article (at least as stated in the title) on p. 6. The rest could also be considered common knowledge.
Line 237: Don’t mitochondria, in general, have reduced DNA repairing capabilities (based on their phylogenetic origin)?
There is a lot of redundancy within many paragraphs (e.e., the role of bioenergetic disturbances in MS pathophysiology is mentioned several times). Please refine the manuscript.
Please add references at the beginning of the discussion. Even though mitochondrial dysfunction has been implicated in many neurological disorders, most clinical trials failed to show any significant therapeutic effect. Obviously, there is no FDA-approved drug for MSS patients targeting mitochondria. The authors should discuss more in-depth what potential causes may be involved in this observation and what would be the next necessary steps.
The authors should also discuss the role of biomarkers of mitochondrial dysfunction in MS patients in more detail.
Author Response
Response to Reviewer 2 Comments
Point 1: Please shorten the introduction and focus on the underlying rationale for this review article.
Response 1: The introduction has been shortened.
Point 2: Please also rephrase the first paragraphs on mitochondrial function. It should be, in general, clear to the readership that mitochondria are involved in, i.e., ATP biosynthesis. What is the precise role of these mechanisms in MS pathophysiology?
Response 2: The information was added at 3.1-3.4§
Point 3: The different sections in the manuscript are somewhat disproportionate. The. The authors start with the main concept of the review article (at least as stated in the title) on p. 6. The rest could also be considered common knowledge.
Response 3: I removed the poorly related paragraph 3.5 and shortened the introduction. Please let me know what else exactly needs to be reduced, if there is a necesserity for it?
Point 4: Line 237: Don’t mitochondria, in general, have reduced DNA repairing capabilities (based on their phylogenetic origin)?
Response 4: Here we meant that mitochondrial DNA mutations are increasing during the MS progression. This does not change the fact that mitochondrial DNA, even in its normal state, has increased instability, which leads to mutations. With MS, this property is enhanced.
Point 5: There is a lot of redundancy within many paragraphs (e.e., the role of bioenergetic disturbances in MS pathophysiology is mentioned several times). Please refine the manuscript.
Response 5: Please point the certain §
Point 6: Please add references at the beginning of the discussion. Even though mitochondrial dysfunction has been implicated in many neurological disorders, most clinical trials failed to show any significant therapeutic effect. Obviously, there is no FDA-approved drug for MSS patients targeting mitochondria. The authors should discuss more in-depth what potential causes may be involved in this observation and what would be the next necessary steps.
Response 6: The information was addded.
Point 7: The authors should also discuss the role of biomarkers of mitochondrial dysfunction in MS patients in more detail.
Response 7: The information was added in discussion.
Round 2
Reviewer 1 Report
Now the manuscript looks more valuable and could be accepted for publication.
Author Response
Response to Reviewer 1 Comments
Point 1: Now the manuscript looks more valuable and could be accepted for publication.
Response 1: Thank you very muсh for the revision.
Reviewer 2 Report
After the additional review, I do not have any further concerns w. r. t. the publication of the article. Only minor English edits are required.
Author Response
Response to Reviewer 2
Point 1: After the additional review, I do not have any further concerns w. r. t. the publication of the article. Only minor English edits are required.
Response 1: A few mistakes have been corrected.